# Targeting the Bet-Hedging Strategy with an Inhibitor of Bacterial Efflux Capacity Enhances Antibiotic Efficiency and Ameliorates Bacterial Persistence In Vitro

**DOI:** 10.3390/microorganisms10101966

**Published:** 2022-10-05

**Authors:** Demosthenes Morales, Sofiya Micheva-Viteva, Samantha Adikari, James Werner, Murray Wolinsky, Elizabeth Hong-Geller, Jinwoo Kim, Iwao Ojima

**Affiliations:** 1Bioscience Division, Los Alamos National Laboratory, Los Alamos, NM 87545, USA; 2Center for Integrated Nanotechnologies, Los Alamos National Laboratory, Los Alamos, NM 87545, USA; 3Institute of Chemical Biology and Drug Discovery, Stony Brook University, Stony Brook, NY 11794, USA

**Keywords:** bacterial persistence, efflux inhibitor, *Burkholderia*, polyketide synthase, histone deacetylase inhibition, antibiotic therapy enhancement

## Abstract

Persistence is a bet-hedging strategy in bacterial populations that increases antibiotic tolerance and leads to the establishment of latent infections. In this study, we demonstrated that a synthetic non-toxic taxane-based reversal agent (tRA), developed as an inhibitor of ABC transporter systems in mammalian cancer cells, enhanced antibiotic killing of persister populations from different pathogens, including *Burkholderia*, *Pseudomonas*, *Francisella*, and *Yersinia*. Acting as an inhibitor of bacterial efflux at 100 nM, tRA99020 enhanced antibiotic efficiency and suppressed the production of natural products of *Burkholderia* species polyketide synthase (PKS) function. We demonstrate that the metabolites produced by PKS in response to stress by different antibiotics act as inhibitors of mammalian histone deacetylase activity and stimulate cell death. Applying a single-molecule fluorescence in situ hybridization (smFISH) assay, we analyzed on a single-cell level the activation profiles of the persistence regulating pks gene in *Burkholderia thailandensis* treated with tRA99020 and antibiotics. We posit that a multi-pronged approach encompassing antibiotic therapies and inhibition of efflux systems and fatty acid catabolism will be required for efficient eradication of persistent bacterial populations.

## 1. Introduction

Phenotypic adaptations that hedge against stressful environmental conditions can fortify bacterial resilience to antimicrobial agents leading to antibiotic tolerance and latent bacterial infections [1,2,3]. Consistent with the bet-hedging strategy, bacterial persisters have lower fitness in prosperous growth conditions (with typically slow division rates) in exchange for increased fitness in stressful conditions, including nutrient limitation and exposure to toxic metabolites and antibiotics [4,5,6,7]. Although bacterial persisters are transiently antibiotic-tolerant and represent a very small fraction of the clonal population of antibiotic-susceptible cells, they greatly contribute to the global antibiotic resistance crisis [8,9].

The clinical impact of bacterial persistence has stimulated fundamental research on elucidation of the molecular mechanisms crucial for bacterial tolerance to antibiotics [10,11]. A remodeling of the lipid component of the bacterial cell walls and the activation of the transmembrane transport systems for an efflux of toxic metabolites have been identified as phenotypic adaptations that increase antibiotic tolerance in various bacterial species [12,13]. Dormant *Mycobacterium tuberculosis* and persistent *Burkholderia thailandensis* cells activate genes regulating fatty acid degradation (*fadE* and *fadD*) [10,14]. Pyrazinamide (PZA) is the only known drug that was found to be effective against dormant *M. tuberculosis*. PZA has been shown to directly inhibit the fatty acid synthase I complex, thus further supporting a role for lipid catabolism in bacterial persistence [14].

Another protein family critical for maintaining bacterial persistence is the ATP-binding cassette (ABC) transporter systems, which promote lipid and amino acid uptake. In *M. tuberculosis*, the Mse4 operon is a cholesterol import system essential for intracellular survival and the establishment of chronic infection [15]. In *B. thailandensis*, several ABC transporter systems are activated in persisters, including lipid and branched-chain amino acid ABC transporters [16]. In addition to their role in metabolic homeostasis, ABC transporter families are part of multidrug efflux systems and support bacterial tolerance to antibiotics by decreasing intracellular drug concentrations [13].

While efflux pump inhibitors, such as PaβN (Phenylalanine-arginine β-naphthylamide) and NMP (1-(1-naphthylmethyl)-piperazine) that target the resistance-nodulation-cell division, an RND-superfamily of efflux pumps, are limited to a small group of bacterial species [17,18], inhibitors of conserved ABC transporter catalytic domains could potentiate antibiotic efficiency against a broader community of bacterial species. To test this hypothesis, we investigated a synthetic, nontoxic taxane-based reversal agent (tRA), a compound initially developed as an inhibitor of multi-drug-resistant ABC transporter families in mammalian cancer cells, including P-glycoprotein (Pgp), MDR protein (MRP-1), and breast cancer resistance protein (BCRP) [19]. Unlike paclitaxel, the tRA compound lacks the C-13 side chain that binds β-tubulin, and therefore, does not arrest cell division [20]. Demonstration of tRA potency against bacterial efflux capacity would suggest the existence of evolutionarily conserved functional domains in ABC transporter systems that could be targeted for the development of broad-spectrum modulators of bacterial persistence.

In this study, our primary model organism for bacterial persistence is *Burkholderia thailandensis*. *Burkholderia* species exhibit inherent resistance to a broad range of antibiotics and tolerance to first-line therapies [21,22,23]. *B. thailandensis* is a well-established surrogate for the etiologic agent of melioidosis, *B. pseudomallei*. Both *Burkholderia* species share common antibiotic resistance mechanisms and an adaptation to host antibacterial responses [24,25]. We have previously identified polyketide synthase (PKS) systems that are significantly upregulated in antibiotic-tolerant populations of *B. thailandensis* and have demonstrated their role in bacterial persistence to antibiotics [10]. Furthermore, these PKS systems have been linked to the production of natural products that inhibit mammalian histone deacetylase (HDAC) function [26]. Here, we demonstrate that antibiotics induce *pks* gene expression in *B. thailandensis*, leading to the secretion of metabolites that subsequently inhibit HDAC function and stimulate apoptosis in human host cells. Additionally, we demonstrate that the inhibition of the bacterial efflux capacity, using a tRA compound, enhanced the antibiotic-mediated killing of several bacterial pathogens, associated with respiratory infections, and ameliorated the cytotoxic effect of the Burkholderia natural products on host immune cells and lung bronchial epithelial cells in vitro tissue cultures.

## 2. Materials and Methods

### 2.1. Bacterial Strains and Reagents

*Burkholderia thailandensis* (*B.th*.) E264 (BEI Resources, Manassas, VA, USA, NR-9907), *B.*
*th.* E264 transposon mutant (#8684) (provided by Dr. Larry Gallagher, University of Washington, Seattle), *Pseudomonas aeruginosa* PA01-LAC (ATCC 47085), and *E. coli* K-12 (ATCC, PTA-4732) cultures were maintained on antibiotic-free LB-Miller agar plates (Fisher Scientific, Cat# BP1425). *Burkholderia pseudomallei* 82 (ϪpurM), a Select Agent exempt pathogen (BEI Resources, NR-51280), was maintained on LB-Millar agar supplemented with 40 µg/mL adenine and 0.0005% thiamine. *Francisella tularensis* subsp *holarctica* CDC Live Vaccine Strain (BEI Resources, NR-646) cultures were maintained on antibiotic-free Chocolate II agar plates (BD, Cat# B21169X). *Yersinia pestis* subsp medievalis KIM5- (pCD1_+_, pgm-), obtained from Dr. Susan Straley’s lab (U. of Kentucky, Lexington, KY, USA) was propagated on brain heart infusion (BHI) agar (Difco, MI). Antibiotics were acquired from Sigma-Aldrich (St. Louis, MO, USA): Cat# M2574 (Meropenem), Cat# 1478108 (Ofloxacin). The tRA99020 drug modulator, provided by Dr. I. Ojima [20], was solubilized in 100% DMSO to prepare stock solutions of 1 mM and 10 mM.

### 2.2. Enumeration of Persister Populations

Bacterial cultures were grown in LB-Miller Broth or MH Broth until culture density reached OD_600_ 0.20–0.30 (mid-exponential phase) and were then exposed to antibiotics (at 1xMBC or higher dose) and/or tRA99020 for 18 h at 37 °C with agitation at 220 rpm. Limiting dilutions (log_10_) of bacterial cultures were plated for counting colony forming units (CFU) before (at 0 h) and after (at 18 h) antibiotic treatment on antibiotic-free agar plates. Persister populations were calculated as a percentage of surviving cells after antibiotic treatment (CFU/mL at 18 h) relative to the number of cells prior to antibiotic treatment.

### 2.3. Evaluation of Bacterial Export Capacity

Planktonic bacteria ware collected at an early exponential phase (up to OD_600_ 0.2) and were pelleted via centrifugation (5 min at 4000 rpm). Bacterial pellets were resuspended in PBS by the adjustment of OD_600_ to 0.1 and Hoechst 33342 (25 mM) was added to each culture. Hoechst 33342—labeled bacterial cultures were distributed in 96-well plates (4 replicas per culture) and recording was initiated 5 min after the addition of the nuclear dye. Excitation and emission (355/460 nm) were measured on BioTeck Synergy H4, utilizing the kinetic mode for readings at 5 min for 14 cycles. Data from each cycle were plotted on Excel. Each experiment was repeated 4 times and the statistical significance was determined with unpaired Student’s *t* test.

### 2.4. Treatment of Human PBMCs and NHBEC with Bacterial Natural Products

Human peripheral blood mononuclear cells (hPBMCs) were treated with the culture supernatants of bacteria grown in mammalian tissue culture media. Human cell preparation: Human peripheral blood mononuclear cells (hPBMCs) were maintained in RPMI-10% medium composed of RPMI (Gibco, Cat# 11835030) supplemented with 10% feteal bovine serum (FBS, ATCC, Cat# 30-2020). HPBMC were treated with 1.5% (*v*/*v*) Phytohemagglutinin M (PHA-M, Gibco, Cat# 10576015) to induce blastogenesis (10 ug per well, Sigma, Cat# F1141-2MG) and were incubated for 72 h at 37 °C with 5% CO_2_. Normal human bronchial epithelial cells (NHBECs) were maintained in BronchiaLife Complete Medium (LifeLine, Cat# LL-0023). Bacterial conditioned media preparation: *B.thailandensis* E264 (WT), *B.thailandensis pks* transposon mutant (#8684), *B.pseudomallei* 82, and *E. coli* K-12 cultures were adapted over 48 h to grow in RPMI 10% media. Antibiotics and/or inhibitor drugs were added to mid-log phase bacterial growth and incubated for 18 h at 37 °C, with agitation at 220 rpm. Treated cultures were pelleted by centrifugation for 5 min at 3000× *g*, and the bacterial media supernatants were filtered using 0.2 µm PES membrane syringe filters (Millipore, Burlington, MA, USA) and stored at 4 °C until needed. Primary human cells were purchased from Lonza Bioscience (Morrisville, NC, USA) and have been classified as an exempt human subject research item.

### 2.5. Histone H3 Acetylation and Caspase 3 Activation Assays

PathScan Acetylated Histone H3 Sandwich ELISA kit (#7232, Cell Signaling Technologies, Danvers, MA, USA) was used to determine changes in acetylated H3 levels in PBMC exposed to bacterial natural products in conditioned media. All ELISA steps were performed according to the manufacturer’s protocol using whole cell lysates from from the following treatment conditions: 10^6^ untreated cells, Trichostatin A (TSA)-treated cells (100 ng/mL), and PBMC incubated in conditioned RPMI-10% media for 24 h. Absorbance readings were taken at 450 nm. Changes in acetylated histone H3 levels were calculated as a ratio of the absorbance at 450 nm for treated samples versus the untreated control sample. The fold change was normalized to the total protein input for each sample. The protein quantity was determined using the RC DC Protein Assay (Cat# 5000122, BioRad Laboratories, Hercules, CA, USA).

The activation of caspase activity in live cells was determined with a fluorescent microscopy assay applying CellEvent Caspase 3/7 Green Detection Reagent at 1 μM concertation (Invitrogen/ThermoFisher Scientific, Waltham, MA, USA). Images were acquired on a Zeiss Axio Observer Z1 microscope.

### 2.6. smFISH Image Analysis

Custom labelled mRNA probes for single molecule Fluorescent In Situ Hybridization (smFISH) assays were obtained from LGC BioSearch Technologies. The custom image analysis program based on Python programming language is available at https://github.com/dmorales003/betaFISH (accessed on 1 August 2022). In brief, DAPI and 16S labeled images were added together to generate a complete reference image of bacterial cells for object segmentation. An adaptive threshold was applied to reference cell images followed by watershed segmentation available through the OpenCV library. mRNA on smFISH labeled images were counted using a Laplacian of Gaussian blob detection algorithm. For spot analysis, only cells with an area within one standard deviation of the mean of the area for each treatment were analyzed. The proportion of active versus non-active cells was determined by bootstrapping, in which the ratio of individual cells with one or more transcripts within 30 random individual cells was determined and repeated 3000 times to acquire a normal distribution of the active proportion. From the normal distribution, datapoints greater or less than 3 standard deviations from the mean were defined as outliers. To measure co-expression, the count of each gene per cell was plotted in a bivariate plot. Histograms and plots of the smFISH data were generated using Pandas in conjunction with Matplotlib and Seaborn libraries.

### 2.7. Statistical Analysis

Statistical significance of the percentage of bacteria surviving antimicrobial therapies, human cells with changes in acetylated histone H3 chromatin levels, and differential gene expression in human cells were determined with the unpaired Student’s *t* test, applying Sidak–Bonferroni method for multiple comparisons when correcting for the *p* values. Statistically significant *p* values were considered when less than 0.05. Microsoft Excel application was applied to analyze the data and to obtain the means and standard errors of the means.

## 3. Results

### 3.1. Taxane-Based Reversal Agent, tRA99020, Acts as an Inhibitor of B. thailandensis Efflux Capacity and Potentiates Antibiotic-Mediated Killing of Bacterial Persister Populations

Here, we investigated the ability of a small molecule compound developed as a broad-spectrum inhibitor of mammalian ATP-binding cassette (ABC) transporter proteins [19] to potentiate antibiotic therapy against *B. thailandensis* persister populations. A cell efflux assay indicated that the taxane-based reversal agent, designated tRA99020, acts as an inhibitor of *B. thailandensis* efflux capacity. Compared to untreated bacterial cultures, *B. thailandensis* cells treated with various concentrations of tRA99020 exhibited increased accumulation of the fluorescent DNA-binding dye, Hoechst 33342 (Figure 1A). We initiated fluorescence intensity (FI) readings at 5 min of incubation with the dye and observed higher Hoechst 33342 retention in tRA99020 treated versus untreated bacterial populations, while the rates of dye accumulation in all tested conditions were similar. These results indicate that tRA99020 had no effect on the dynamics of dye entry into the cells, while it strictly inhibited Hoechst 33342 export. Furthermore, with this method we did not observe a concentration dependent inhibition of bacterial efflux capacity. This could be pertinent to the limitations in the method sensitivity or mechanisms of small molecule action that is based on a threshold effect (“all or nothing”). To evaluate the statistical significance of Hoechst 33342 retention upon tRA99020 treatment, we used the FI data from steady state dye accumulation originating from four independent bacterial culture experiments (Figure 1A, bar figure). The dynamics of Hoechst 33342 accumulation in tRA99020 treated *B. thailandensis* versus untreated bacterial populations are comparable to previously reported efflux pump knockout mutant (Δ*glu*P) of *H. pylori* [27]. Previous studies have demonstrated that the bacterial MDR exporters, Sav1866 and MsbA, together with their mammalian functional ABC transporter homologs (ABCB1 and ABCG2) share transport substrate specificity that includes Hoescht 33342 [28,29,30]. Thus, retention of Hoescht 33342 by tRA99020-treated *B. thailandensis* cultures suggests that the small-molecule compound may target evolutionarily conserved ABC transporter systems in bacteria.

Next, we supplemented antibiotic treatment with the tRA99020 compound and observed that the antibiotic potency was amplified (≥10-fold) when the ABC transporter inhibitor was applied at 100 nM concertation. Quite unexpectedly, at 1 µM concentration the effect of the efflux inhibitor tRA99020 took a reverse course of action, as we have consistently registered obstruction of the antibiotic bactericidal efficiency (Figure 1B). This contrasting effect was more pertinent to *B. thailandensis* treatment with ofloxacin (Ofl) compared to supplementation of meropenem (Mpm) with 1 µM tRA99020 (Figure 1B). Importantly, tRA99020 did not act as an inhibitor of bacterial growth (Figure 1B) and was able to potentiate antibiotic activity at 1 µM against persister populations of various Gram-negative bacterial species, including *Francisella tularensis*, *Pseudomonas aeruginosa*, and *Yersinia pestis* KIM5 (Figure 2). For these bacterial species, the reverse effect of tRA99020 on antibiotic bactericidal capacity was observed between 10 and 20 µM (data not shown). Our results demonstrate the efficacy of this small molecule as a broad-spectrum modulator of bacterial persistence and emphasize the differences in sensitivity towards the tRA99020 compound among bacterial species.

### 3.2. At 1 µM Concentration tRA99020 Stimulates Expression of Bacterial Persistence pks Gene in B. thailandensis Populations

We further investigated tRA99020 dosage effect on the activation of a gene associated with antibiotic tolerance in *B. thailandensis* persister cells. In our previous study, using comparative transcriptomics and functional genomic assays, we have demonstrated that polyketide synthase (PKS) function played a significant role in *B. thailandensis* persistence to a variety of antibiotics [10]. Here, we utilized the *pks* gene expression as a biomarker of persistence onset. Given the stochasticity and the rarity of persister cell occurrences in bacterial populations (0.1–10%), we studied persistence gene activation via smFISH counts of mRNA copies in singular cells (Figure 3A). We observed an increase in the population of *B. thailandensis* cells with activated *pks* gene transcription in the stationary growth phase and upon treatment with antibiotics (Figure 3B). Compared to untreated bacteria, exponentially dividing *B. thailandensis* cultures treated with a sub-micromolar tRA99020 concentration (100 nM) showed up to a 15% decrease in *pks* transcription (Log, “−” vs. “+”). Relative to the samples treated only with antibiotic, tRA99020 added to an antibiotic media at 100 nM reduced the population of *B. thailandensis* cells expressing *pks* (“−” vs. “+”) at least by 10%. Exposure of log *B. thailandensis* populations to 1 μM tRA99020 resulted in an increased percentage of bacterial cells expressing *pks* (“+” vs. “++”) (Figure 3B, upper panel, log). In a separate study, we exposed logarithmically expanding *B. thailandensis* cultures to 1 μM tRA99020 and detected a 50% increase in the population of *pks* expressing cells solely in the presence of the efflux inhibitor, while a combination treatment of tRA99020 (1 μM) with ofloxacin caused an additional 10% increase of the persistence gene expressing population when compared to the pks gene activation caused by the singular application of the antibiotic (Figure 3B, lower panel). Notably, even a 1% increase in the population of bacterial cells expressing the persistence regulating *pks* gene will have a significant impact on the microbicidal potency of the antibiotics that would leave behind hundreds of live cells to rekindle the infection upon antibiotic withdraw.

These findings indicate that at higher concentrations this efflux inhibitor stimulated expression of the bacterial persistence gene *pks*, and thus may account for the decreased antibiotic efficiency observed in our cell survival assay (Figure 1B).

Next, we investigated the co-expression levels of *fad*E and *pks* genes in *B. thailandensis* populations under various stress conditions, including treatment with tRA99020. FadE (BTH_RS10725) is a key enzyme in the β-oxidation of long-chain fatty acid pathway which t provides metabolites for PKS function (Appendix A). With this study we aimed to investigate the impact of the small molecule inhibitor of bacterial efflux capacity on the persistence metabolic program by observing the gene expression profiles of enzymes which function is biochemically linked. We utilized spectrally distinguishable smFISH probes for *fadE* and *pks*. Dot plots, representing gene mRNA counts per single *B. thailandensis* cell, demonstrated that Ofl could stimulate co-expression of *fade* and *pks* genes, whereas Mpm triggered only *fadE* gene activation (Figure 3C). Furthermore, supplementation of the antibiotic treatment with 0.1 μM tRA99020 specifically inhibited *pks* gene activation in Ofl-treated populations, but had no effect on *fad*E gene activation in antibiotic-treated *B. thailandensis* cells (Figure 3B). This finding suggested a role for fatty acid catabolism in the establishment of the persistence metabolic program. We provide further support to this idea with an experiment wherein Ofl treatment combined with an inhibitor of fatty acid amide hydrolase (FAAH) activity resulted in a 50% reduction of bacterial survival rates compared Ofl treatment alone (Figure 4).

Collectively, the data presented herein indicate that, at a sub-micromolar concentration, tRA99020 could potentiate an antibiotic microbicidal effect by inhibiting the expression of the persistence supporting *pks* gene. In contrast, the micromolar concentration of the same compound stimulated this persistence gene in *B. thailandensis* cells. Co-activation of *fadE* and *pks* suggests that catabolism of long-chain lipids may contribute to the onset of persistence in response to antibiotics. Due to the variety of overlapping mechanisms contributing to the bacterial antibiotic tolerance, we found that tRA99020 ameliorated, but did not abolish, the bacterial persistence in response to antibiotics. Since tRA99020 did not activate *fade*, we conclude that the small molecule does not affect gene transcription in general, and thus has an indirect effect on the stimulation of *pks*, presumably by inhibiting bacterial efflux capacity and disrupting metabolite homeostasis the compound contributed to establishment of stressful conditions in the microbial cells.

### 3.3. Natural Products of Burkholderia PKS Function Modulate HDAC Activity in Live Human Peripheral Blood Mononuclear Cells

We have previously shown that polyketide metabolites produced by *B. thailandensis* PKS systems act as inhibitors of mammalian histone deacetylase (HDACi) enzymes in in vitro assays [10]. Since HDAC activity in mammalian cells is essential for rapid immune responses to infections [31,32], we tasked ourselves with testing the efficacy of *B*. *thailandensis* natural products as modulators of HDAC activity in live host cells. Here, we utilized ELISA to quantitatively evaluate the levels of acetylated histone H3 in PBMC cultures exposed to conditioned media from two *Burkholderia* species, *B. thailandensis* E264 and *B. pseudomallei* 82, and the opportunistic pathogen *P. aeruginosa*. We found that natural products released into the conditioned media of *Burkholderia* spp. increased the levels of acetylated H3 in human PBMC cultures similar to treatment with the synthetic HDACi Trichostatin A (Figure 5A, TSA). Conditioned media from ofloxacin treated *B. p**seudomallei* 82 cultures did not show potency as HDACi, most likely due to the reduced counts of live bacteria. In contrast, HDACi efficacy remained unchanged in Ofl-treated *B. tailandensis* E264 cultures relative to untreated cultures. These results suggest that the genes responsible for the HDACi natural products have been significantly activated in the *B. thailandensis* E264 persister cells that survived antibiotic treatment, thus compensating for the reduction of live cell counts caused by the antibiotic. Conditioned media obtained from *P. a**eruginosa* cultures did not exhibit HDACi activity. Notably, the PKS functional mutant strain *B. thailandensis* 8684 *(B.th* m*86), did not produce metabolites that altered the levels of acetylated H3; thus, further validating the role of PKS products as regulators of both bacterial resilience to antibiotics [10] and host responses to bacterial infection through epigenetic modifications (Figure 5A). Collectively, these findings demonstrate that *Burkholderia* spp. produce metabolites that modulate histone acetylation and thus have the propensity to alter gene expression profiles in host immune cells.

### 3.4. Natural Products of B. thailandensis PKS Function Inhibit Pro-Survival Mechanisms in Host Cells

Upon exposure of human PBMCs to bacterial conditioned media (for the HDACi assay), we noticed that natural metabolites released into *B. thailandensis* E264 cultures exhibited pronounced toxicity to the mammalian cells. The cytotoxic effect of *B. thailandensis* E264 conditioned media was not limited to the human PMBC, as we also observed massive cell death of normal human bronchial epithelial cells (NHBEC) upon 24 h of exposure to the *B. thailandensis* natural products (Figure 5B). In stark contrast, conditioned media from the *B. thailandensis pks* null mutant strain 8684 (*B.th*. m*86) did not exhibit pronounced cytotoxicity (Figure 5B, lower panel). This finding prompted further investigation of the expression profiles of genes regulating cellular pro-survival responses (CFLAR) and cell death (Casp3) in hPBMC exposed to filter sterilized media from *B. thailandensis* cultures, wild type E264 strain (*B.th.* E264,-WT), and the PKS mutant strain (*B.th._*m*86). We included exotoxins from *E. coli* as controls, both purified lipopolysaccharide (LPS) and raw metabolites released by replicating *E. coli* K12 cultures. Caspase 3 and FADD-like apoptosis regulator (CFLAR), an inhibitor of apoptosis, have a crucial role in the regulation of cellular death by apoptosis and necrosis [33]. We observed a significant activation of *casp3* gene expression in hPBMC cultures exposed to conditioned media from wild type *B. thailandensis* E264 (Figure 5C, *B.th.* E264, black bars) compared to media from *B. thailandensis* Ϫ*pks* mutant bacteria (Figure 5C, *B.th.* m*86, black bars). Notably, *B.th*_E264 conditioned media strongly inhibited the pro-survival *cflar* gene expression in immune cells, while metabolites released from the PKS mutant strain activated *cflar*. In comparison, LPS treatment and conditioned media from antibiotic-free *E. coli* K12 cultures had no detectable effect on *casp3* or *cflar* gene activity (Figure 5C). Furthermore, antibiotic treatment of the *B. thailandensis* Ϫ*pks* mutant cultures increased the cytotoxic effect caused by the released bacterial metabolites and triggered the activation of both *casp*3 and *cflar* genes. Wild type *B.th*_E264 cultures treated with antibiotics released metabolites that stimulated the apoptotic casp3 gene, but they retained their inhibitory effect on *cflar* gene expression, similar to antibiotic free cultures. Collectively, our data strongly indicate that the metabolite synthesized by the BTH_RS24270-encoded PKS function inhibits host cell pro-survival mechanisms and activates apoptosis in mammalian cells.

### 3.5. Supplementation of Antibiotic Treatment with tRA99020 Agent Ameliorates the Cytotoxic Effect of Burkholderia spp. Natural Products on Host Cells

We further examined how the inhibition of bacterial efflux activity by tRA99020 would affect host cell survival responses to metabolites released by bacteria upon antibiotic treatment. In this study, we used the Select Agent exempt pathogen *B. pseudomallei 82* (*B.p. 82*) harboring intact PKS systems, and two opportunistic pathogens: *B. thailandensis* E264 (*B.th.* E264) and *P. aeruginosa* (P.a.). Conditioned media from antibiotic-free *B. pseudomallei* 82 cultures stimulated the pro-survival *cflar* gene in hPBMCs but had no effect on *casp*3 gene expression in these cells (Figure 6A, black bars). *P. a**eruginosa*-secreted metabolites affected neither gene activity, while exposure to *B. thailandensis* metabolites inhibited the expression of *cflar* and activated *casp*3 by two-fold compared to untreated hPBMCs (Figure 6A). We found that antibiotic treatment of *Burkholderia* spp. increased the release of cytotoxic metabolites into the culture media as evidenced by the stimulation of *casp*3 gene expression in in exposed hPBMC (Figure 6A, grey bars). Importantly, supplementation of ofloxacin with 0.1 μM of the bacterial efflux inhibitor, tRA99020, reduced the cytotoxic effect of bacterial metabolites, as we observed a decrease in *casp*3 gene activity in the host immune cells (Figure 6A, white bars). This coincided with restoration of the *cflar* gene expression in hPBMCs exposed to *B. thailandensis* conditioned media (Figure 6A, white bars).

We observed a similar effect of the bacterial metabolites on cell death activation in normal human lung bronchial epithelial cells (NHBECs). We exposed NHBEC to filter-sterilized conditioned media from *B. thailandensis* E264, *B. thailandensis* *m86, and *B. p**seudomallei* 82 cultures treated with antibiotics, and compared the host cell pro-survival responses to that of cells exposed to bacterial conditioned media without antibiotics (Figure 6B). Gene expression of the pro-survival CFLAR host factor was inhibited in NHBEC exposed to bacterial metabolites secreted by *B. thailandensis* E264 and *B. pseudomallei* 82, in response to antibiotic treatment compared to treatment with conditioned media from the *pks* mutant *B. thailandensis* (*B.th.* m*86) and antibiotic-free *B. pseudomallei* cultures. In reciprocal correlation, we found that the *casp3* gene was activated NHBECs exposed to metabolites from all three antibiotic treated *Burkholderia* cultures: s, *B.th*. E264, *B.th*. m*86, and *B.p*. 82. Collectively, data presented here solidified our findings that antibiotic stress stimulates the secretion of bacterial metabolites by *Burkholderia* spp. that in turn manipulate host cell survival responses and stimulate apoptosis. The bacterial efflux inhibitor, tRA99020, when applied at a submicromolar concentration reduced *pks* gene expression in *B. thailandensis* cultures (Figure 3B) and was able to ameliorate the cytotoxic effect of *Burkholderia* spp. metabolites on host cells (Figure 6A).

## 4. Discussion

Bacterial persistence is a phenomenon where transient phenotypic changes, often induced by environmental stressors, increase bacterial tolerance to cytotoxic antibiotics. Multiple mechanisms have been identified to drive bacterial persistence, including stringent response to nutrient starvation and toxin-antitoxin (TA) systems regulating protein translation, DNA stability, and cell membrane integrity [34,35,36].

Our previous study has identified ABC transporters, genes regulating lipid catabolism (specifically β-oxidation), and PKS enzymes (involved in the synthesis of polyketide secondary metabolites) to be significantly activated in *B. thailandensis* populations surviving antibiotic treatment [10]. Here, we provide evidence of co-activation of the persistence gene *pks* and *fad*E, a key regulator of long-chain fatty acid β-oxidation [37], in a single bacterial cell in response to antibiotics and a small-molecule inhibitor of bacterial efflux capacity. We observed shifts in the overall size of bacterial populations expressing the *pks* gene, whichcould explain the divergent mechanisms of bacterial persistence in response to various classes of antibiotics and different concentrations of the compound inhibitinf bacterial efflux capacity.

ABC transporter systems are excellent anti-persistence drug target candidates due to their evolutionarily conserved role in the maintenance of metabolite homeostasis. ABC transporters are found in all domains of life as part of a large gene family that supports translocation of various substrates across biological membranes. These substrate transporter systems consist of conserved nucleotide-binding domains (NBDs) that hydrolyze ATP and induce conformational changes in the transmembrane domains (TMDs) to regulate transport of specific substrates across lipid membranes [38]. In bacteria, ABC transporters facilitate the uptake of nutrients and the export of building blocks for cell-wall assembly [39]. Substrate-promiscuous ABC systems that export various toxic substances out of the bacterial cell contribute to multi-drug tolerance [40]. Similarly, mammalian multi-drug ABC transporters are essential for the protection of various organ tissues from toxic compound accumulation and largely contribute to chemotherapy tolerance of cancer cells [41]. Despite a moderate (up to 50%) sequence similarity within their NBDs and a low (up to15%) TMD sequence similarity, the mammalian and bacterial ABC transporters share specificity for multiple substrates [42]. This motivated our current study where we tested a small-molecule compound, tRA99020, for its possible application as a broad-spectrum modulator of persistence in bacteria. Our finding that a small-molecule drug developed initially as a modulator of mammalian ABC transporters with multi-drug resistant (MDR) function [19] could inhibit substrate export by bacterial cells (Figure 1A) signifies that conserved functional domains in ABC transporter systems may present viable targets for the development of broad-spectrum modulators of bacterial persistence.

Supplementation of antibiotic treatment with the tRA99020 compound amplified the antibiotic bactericidal effect against *B. thailandensis* (≥10-fold) when applied at 100 nM concentrations (Figure 1B). On the contrary, concentrations ≥1 µM tRA99020 consistently reduced antibiotic potency against *B. thailandensis* persister populations. We examined the possible mechanisms for this functional divergence by observing the effect of tRA99020 on the gene activation profiles of *B. thailandensis* persistence biomarkers pks and *fadE* at a single-cell level via smFISH. Our previous study has shown that these genes were upregulated in *B. thailandensis* populations surviving antibiotic treatment [10]. FadE is a key enzyme in the β-oxidation of long-chain lipids that provides building blocks for the PKS enzymes involved in the synthesis of polyketide secondary metabolites and modulators of bacterial tolerance to antibiotics (Appendix A) [43,44]. Our data demonstrated that the persistence-regulating gene *pks* was stimulated at a higher frequency in antibiotic-tolerant cells, while not completely turned off in the antibiotic-naïve populations (untreated *B. thailandensis*). We also observed that meropenem, which targets bacterial cell wall synthesis, significantly expanded the population of cells with activated *fadE* gene transcription, while ofloxacin, an inhibitor of DNA replication, stimulated co-expression of *fadE* and *pks* genes (Figure 3C).

The smFISH data further solidified our understanding that, at nM concentrations, tRA99020 lowered *B. thailandensis* resilience to antibiotic killing (Figure 1B) by reducing *pks* gene expression with bacterial populations (Figure 3B). Importantly, in mammalian cancer cells, the effective dose of this tRA compound was 10 µM [45], indicating that the small-molecule drug could be safely repurposed in nM amounts as a supplement to antibiotic therapy with minimal on mammalian host function. Additionally, our findings indicate that tRA99020 indirectly affects persister gene expression by creating stressful conditions in bacterial cells, and that the use of drug efflux inhibitor alone would not be sufficient for the effective eradication of the *B. thailandensis* persister populations. Based on data presented here, a combination therapy consisting of antibiotics supplemented with inhibitors of bacterial efflux capacity and lipid catabolism would constitute a more potent anti-persistence therapeutic formulation.

We identified the natural product of *B. thailandensis* PKS function to inhibit mammalian HDAC activity, thus affecting Caspase 3 and FADD-like apoptosis regulator (CFLAR) transcription in mammalian host cells, and accounting for the observed enhanced toxicity of bacterial metabolites. We found a strong association between *pks* gene activity in bacteria and the cell death of host immune and lung cells. Filtered conditioned media from wild-type *B. thailandensis* promoted the expression of the pro-apoptotic enzyme Casp3 in hPBMCs, but inhibited the expression of the apoptosis regulator, CFLAR (Figure 5 and Figure 6). We observed similar gene expression dynamics in human lung cells in response to *Burkholderia* metabolites exposure (Figure 6B). As conditioned media from the PKS functional mutant (*B.th.*m*86) did not inhibit *cflar* gene expression (Figure 5C), we posit that the polyketide natural products have an inhibitory effect on host cell survival mechanisms. In this study, we have shown that antibiotic-treated *B. thailandensis* express more frequently the *pks* gene and release metabolites that are more toxic to the host immune and lung cells than antibiotic-naïve populations. Supplementation of antibiotic treatment with tRA9920 inhibited *pks* gene expression in *B. thailandensis* and resulted in increased *cflar* gene expression in the host cells exposed to bacterial metabolites.

Finally, we have demonstrated that natural products from both *Burkholderia* species enhanced the acetylation levels on histone H3 proteins in human PBMCs. This is consistent with previous findings that the natural product of PKS has the properties of a HDAC inhibitor [26]. Given the role of HDAC inhibitors as suppressors of host immune response to microbial infections [31,46], our study reveals a novel mechanism for the antibiotic-induced establishment of persistent bacterial infections. Multi-omics studies can enhance our comprehension of the mechanisms of bacterial transient antibiotic resistance [47]. To push the envelope, here, we provided a functional genomics study supplemented with in situ single-cell analysis of persistent gene activation; however, further proteomics and metabolomics analyses are needed for a clear understanding of the direct link between regulatory pathways responsible for transient antibiotic tolerance and resistance [48,49,50]. Considering the small size of the bacterial persister populations, current advances in single-cell label-free proteomics techniques [51] provide powerful platforms for such an endeavor.

## 5. Conclusions

Antibiotics are frontline countermeasures to infectious diseases caused by pathogenic bacteria. While killing large fractions of susceptible bacterial populations (90–99%), they are proving ineffective against small populations of transiently drug-resistant bacteria, designated as persisters, which give rise to chronic infections. To complicate the issue further, the antibiotics stimulate genes that fortify bacterial stress adaptation mechanisms, which in turn cause broad antibiotic tolerance. The significance of our research is in demonstrating the role of primary and secondary biosynthetic pathways in the establishment of the persistence phenotype and their co-dependence on bacterial efflux capacity. Our finding is imperative t for the identification of evolutionarily conserved targets that will advance the development of enhanced antimicrobial therapies.

## Figures and Tables

**Figure 1 microorganisms-10-01966-f001:**
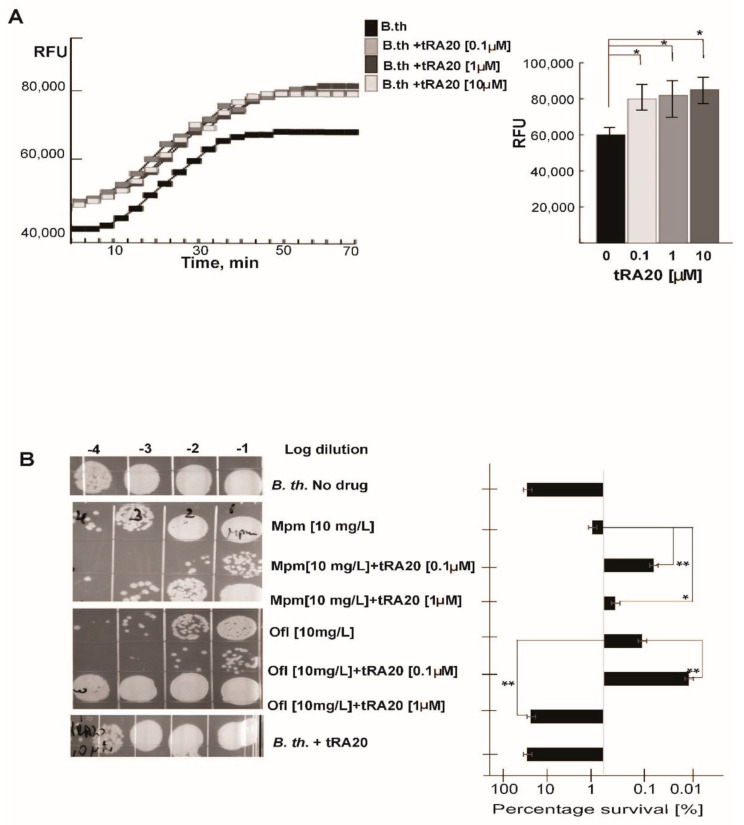
Reduction of *B. thailandensis* efflux capacity by a broad-spectrum inhibitor of mammalian ABC transporter systems affects the rate of bacterial persistence. (**A**) Accumulation of Hoechst 33342 (25 mM) was recorded as fluorescence intensity (RFU, relative fluorescence unites, Ex/Em 350/460) at 5 min after dye addition in untreated or pretreated with tRA99020 planktonic *B.th.* cultures. The kinetic of dye accumulation reflects 3 min recording intervals over a 60 min period. Data shown in bars represent the average and ± standard errors of the steady-state RFU means from four independent bacterial cultures. Paired Student’s *t* test was applied to calculate the significance (*, *p* < 0.01) of Hoechst 33342 accumulation in tRA99020-treated versus untreated bacterial cultures; (**B**) persistence was calculated as a percentage of surviving *B.th*. cells after 24 h of exposure to microbicidal concentrations of antibiotics relative to the bacterial cell number calculated prior to the addition of antibiotic. Bacterial counts at the early exponential growth curve (OD600 0.2) and after antibiotic treatment was determined as colony forming units at log10 serial dilutions. The tRA9902 compound was added at the indicated concentrations in combination with the antibiotics meropenem (Mpm) and ofloxacin (Ofl). The bar graph shows the average and ± standard errors of the means from six biological replicas. The statistical significance (*, *p* < 0.01 and ** *p* < 0.001) of tRA99020’s effect on antibiotic bactericidal efficiency was determined with a paired Student’s *t* test.

**Figure 2 microorganisms-10-01966-f002:**
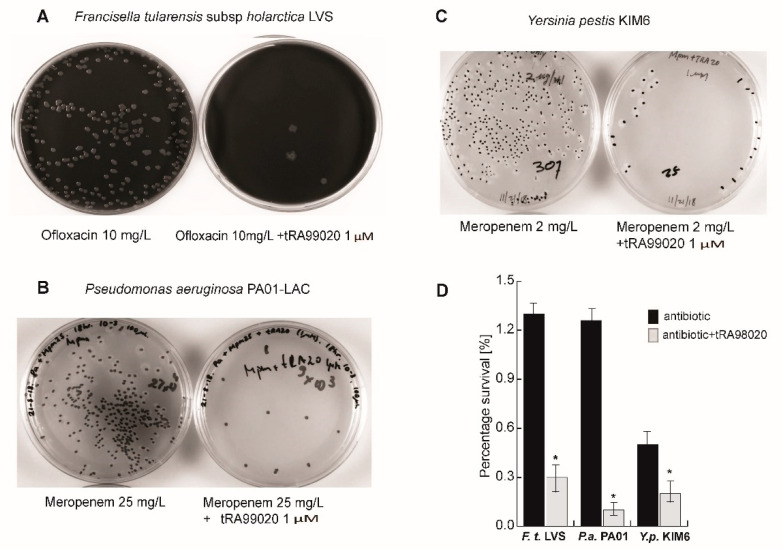
tRA99020 acts as broad-spectrum modulator of bacterial persistence. Bacterial cultures grown to early exponential phase (OD 0.2) were split to equal volumes and antibiotic alone or in combination with tRA9920 compound (1 μM) were added. Upon incubation at 37 °C on antibiotic-free agar plates, the percentage of bacterial survival was determined via limiting dilutions of bacterial cultures relative to the bacterial counts prior to drug treatment. (**A**) Shown are *F. tularensis* LVS colony forming units on chocolate blood agar plates at equal log_10_ (2) dilution rates after 72 h of bacterial growth recovery; (**B**) recovery of *P. aeruginosa* PA01-LAC persisters post-antibiotic treatment was evaluated on LB (Miller) agar plates at 24 h of incubation; (**C**) the effect of tRA99020 on *Y. pestis* KIM6 persistence was evaluated on brain heart infusion agar at 48 h of incubation; (**D**) the average and standard deviations of bacterial survival rates from four independent experiments are shown. For each experiment, the CFU/mL was determined from three technical replicates of colony counts from agar plates with dilution rates resulting in clear separation of individual colonies (10–100 CFU). The significance (*, *p* < 0.01) of tRA99020 effect was calculated with a Student’s *t* test.

**Figure 3 microorganisms-10-01966-f003:**
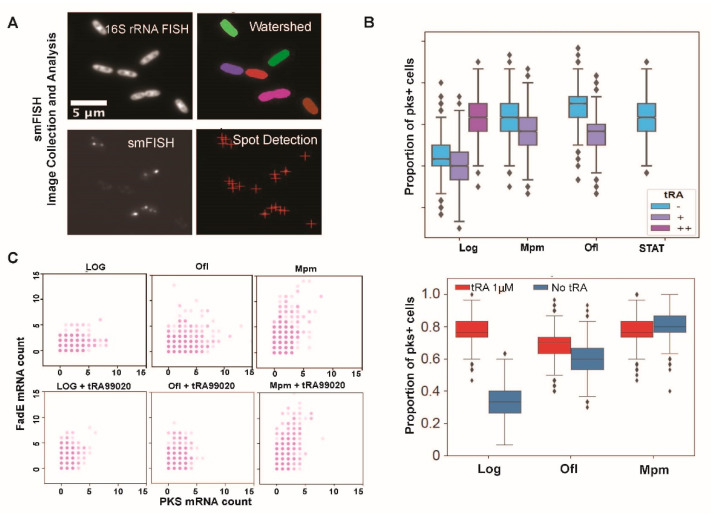
Analysis of persistence *pks* gene expression in *B. thailandensis* E264 populations on the single-cell level via smFISH in response to tRA99020 and antibiotic treatment. (**A**) Fluorescent microscopy imaging of 16S rRNA FISH probes were used for watershed segmentation and smFISH probes for spot detection of PKS mRNA; (**B**) quantitative analysis of smFISH samples, targeting PKS (BTH_RS24270) transcripts in *B.th*. cultures, was performed at various growth conditions (logarithmically expanding, Log, and stationary phase growth, STAT) and antibiotic treatments (meropenem, Mpm [10 mg/L], and ofloxacin, Ofl [10 mg/L]). The efflux inhibitor tRA99020 was added at 0.1 μM (+) as a single treatment or complimentary to antibiotic therapy and at 1 μM (++) as mono-treatment of log cultures (upper panel). The lower panel shows the effect of 1 μM tRA99020 on *pks* gene activation in exponentially expanding population with and without antibiotics. The proportion of cells with target gene activation were determined by bootstrapping (statistical analysis described in the Materials and Methods). Shown are histograms of the ratio of cell populations with one or more transcripts over the total number of cells in the population repeated 3000 times. Outliers, determined as datapoints greater or less than three standard deviations from the mean, are shown as diamonds. Notches on the boxplots describe the 95% confidence interval and non-overlapping notches provide evidence of statistically significant difference of medians (*p* < 0.05, corrected with Sidak–Bonferroni method for multiple comparisons). Each plot shows a representative of three independent experiments; (**C**) smFISH co-expression analysis of *fad*E and *pks* in *B.th*. E264 with spectrally separated Quasar570 and Cal Fluor Red 610-labeled probes. mRNA counts were determined as single “spots” emitted from each gene-specific probe. Dot plots present gene transcript counts per cell from a representative of three independent biological replicas.

**Figure 4 microorganisms-10-01966-f004:**
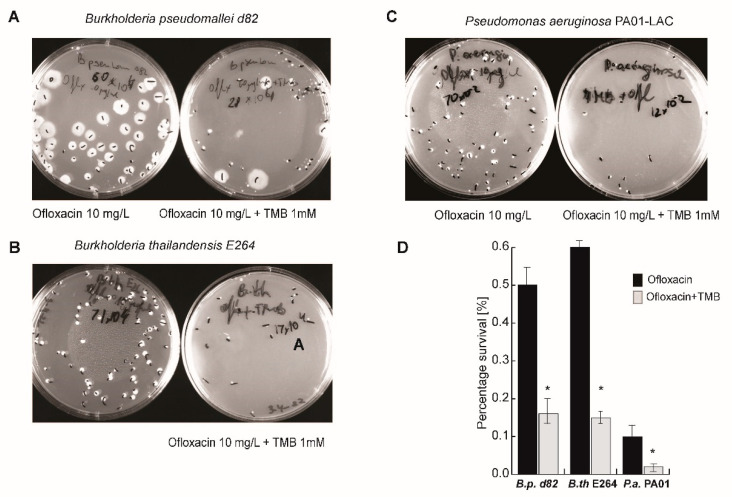
Fatty acid regulatory pathways contribute to bacterial persistence. Thrimethilbenzothiazole/piperazine (TMB), an inhibitor of fatty acid amide hydrolase activity (FAAH), was applied to supplement ofloxacin treatment. Early exponential phase (OD 0.2) bacterial cultures were treated with ofloxacin or with TMB (1 mM) and ofloxacin for 24 h (at 37 °C). Shown is the percentage survival of (**A**) *B*. *pseudomallei* d82; (**B**) *B. thailandensis*; and (**C**) *P. aeruginosa* PA01-LAC cultures calculated as ratio of colony forming units (CFU) in bacterial populations treated with antibiotic/s relative to CFU counts prior to drug exposure; (**D**) average and standard deviations were calculated from four biological experiments and statistical significance (* *p* < 0.01) was determined with an unpaired Student’s *t* test.

**Figure 5 microorganisms-10-01966-f005:**
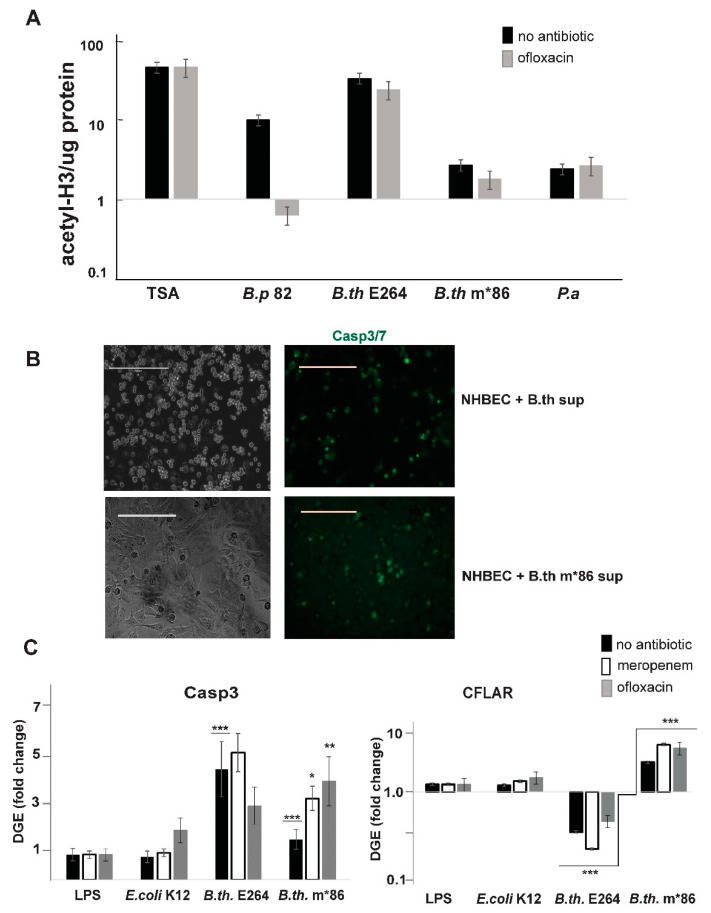
The natural product of PKS (BTH_RS24270) function in *B. thailandensis* (burkholdac) inhibits histone deacetylase (HDAC) function and CFLAR-regulated pro-survival mechanism in host peripheral mononuclear blood cells (PBMC). (**A**) ELISA specific to acetylated histone 3 was performed on nuclear extracts from PBMC exposed for 24 h to conditioned media from naïve or antibiotic-treated *B. pseudomallei* Δ82 (*B.p*. 82), *B. thailandensis* E264 (*B.th.* E264), PKS mutant *B.th.* #8684 strain (*B.th*. *m86), and *P. aeruginosa* (*P.a*.). The absorbance at OD450 was normalized to total input protein per sample. Trichostatin A (TSA) treatment (100 ng/mL) was used as a positive control for HDACi function. The mean and standard deviations were obtained from four technical replicas of a representative of three independent PBMC exposure experiments; (**B**) micrographs of NHBEC after 24 h exposure to filtered conditioned RPMI-10% media from *B.th.* E264 and *B.th*. m*86 (*pks* mutant) bacterial strains. Brightfield microscopic images of NHBEC were matched with fluorescent images from cells labeled with CellEvent Caspase-3/7 green detection reagent. The white scale bars correspond to 100 μm; (**C**) RT-qPCR analysis of differential gene expression (DGE) in PBMC in response to bacterial conditioned media filtrate. DGE was determined as fold change (2^−ΔΔC^) relative to mock-treated PBMC. The Ct signals from the Taqman assay for each gene were normalized to GAPDH. Shown are the means and ± standard errors of the means from three independent experiments performed with batch of PBMC exposed to filtrates from three independent *B. thailandensis* and *E. coli* K14 growth conditions. Filtrates from RPMI medium supplemented with 10% bovine serum with or without antibiotics (meropenem, Mpm [10 mg/mL], and ofloxacin, Ofl [10 mg/mL]) served as mock treatment controls to filtrates from 24 h bacterial cultures in RPMI-10% untreated or treated with antibiotics. Purified LPS from *E. coli* was added at 1 mg/mL to the RPMI media. PKS mutant *B. thailandensis* #8684 strain (B.th. *m86) was used as control for the burkholdac effect (product of PKS BTH_RS24270) on host cell survival. Statistical significance (* *p* < 0.05; ** *p* < 0.01; *** *p* < 0.001) was determined with an unpaired Student’s *t* test from the average and standard deviations from four biological experiments.

**Figure 6 microorganisms-10-01966-f006:**
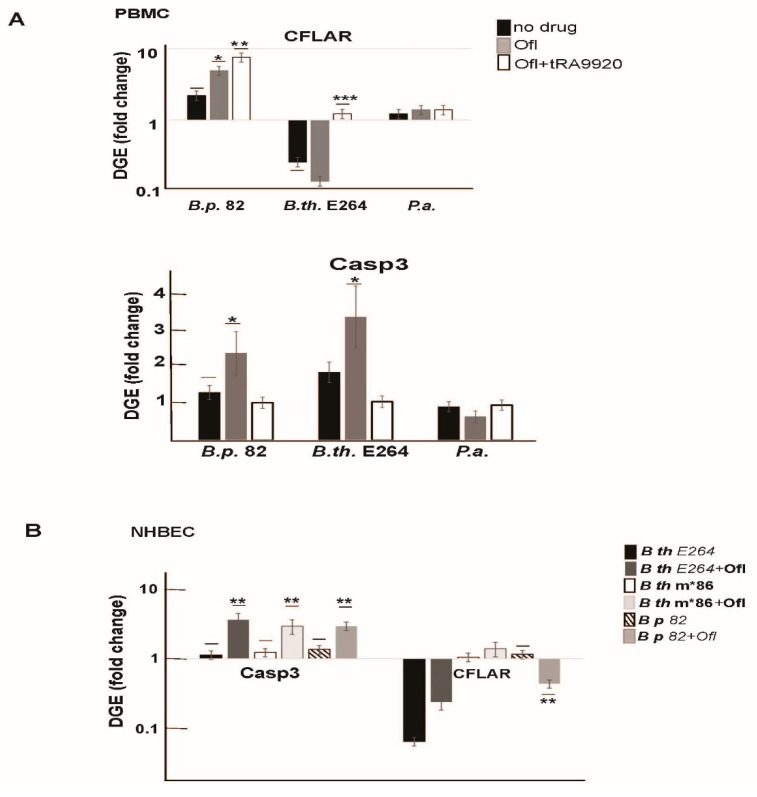
Antibiotic treatment exacerbates the cytotoxic effect from Burkholderia natural products and inhibition of bacterial efflux capacity with tRA99020 counteracts this effect by restoring CFLAR gene expression in host cells. (**A**) RT-qPCR analysis of cell-survival (CFLAR), and apoptosis (Casp3) regulating genes in PBMC exposed to filtered conditioned media (RPMI-10%) from untreated or antibiotic (Ofl)-treated bacterial cultures. DGE in PBMC exposed to natural products of *Burkholderia pseudomallei* Δ82 (B.p. 82), *B. thailandensis* E264 (B.th. E264), and *P. aeruginosa* (P.a.), released in the media upon indicated drug treatment regiments, were calculated relative to gene transcript levels in mock-treated PBMC, including drug-free RPMI-10% and media containing Ofl or Ofl supplemented with 0.1 μM tRA99020. Data show the means and standard deviations from three independent exposure experiments using a PBMC batch from the same donor; (**B**) RT-qPCR analysis of CFLAR and Casp3 DGE in normal human bronchial epithelium cells (NHBEC) exposed to conditioned media from untreated or antibiotic (Ofl)-treated *Burkholderia* species. The DGE was calculated relative to gene transcript levels in mock-treated NHBEC from three independent biological experiments. GAPDH transcripts were used as an internal loading control. Unpaired Student’s t test with Sidak–Bonferroni method for multiple comparisons were applied to correct for the p values. Results were considered statistically significant at * *p* < 0.05, ** *p* < 0.01, and *** *p* < 0.001.

## Data Availability

Not applicable.

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
