# Peer review of "Targeting the Bet-Hedging Strategy with an Inhibitor of Bacterial Efflux Capacity Enhances Antibiotic Efficiency and Ameliorates Bacterial Persistence In Vitro"

_microorganisms, 2022, doi:10.3390/microorganisms10101966_

Round 1
Reviewer 1 Report
In the paper, the authors have described that an ABC transporter inhibitor, tRA99020 (tRA20), could effectively enhance the efficacy of antibiotics and prevent bacteria persistence. The findings are interesting and the study design is sound, which included human PBMC experiments. Below are some concerns of mine.
1. In Figure 1B and related text, tRA20 showed expected effect at 0.1uM but reverse effect at 1uM. The authors did give enough discussion about this issue and use 1uM in subsequent experiments (Figure 2). And how were the colony forming units counted as they seem to merge together in some experiments?
2. Figure 2 should be separated into four panels.
3. Figure 3B has no information about the statstical significance (p-values).
4. Label-Free and single cell proteomics are the leading techiques in analyzing antibiotic resistance (Expert Review of Proteomics, 2019, 16(10):829-839), and recent technical advances can be discussed (Analytical Chemistry, 2022, 94(15):6026-6035; Microsystems & Nanoengineering, 2022, 8:13).
5. Authors of reference #34 is not anonymous, check carefully.
6. The layout of Figure 6 should be rearranged to avoid large blank.
7. Check the manuscript carefully for redundant spaces between words.
Author Response
- In Figure 1B and related text, tRA20 showed expected effect at 0.1uM but reverse effect at 1uM. The authors did give enough discussion about this issue and use 1uM in subsequent experiments (Figure 2). And how were the colony forming units counted as they seem to merge together in some experiments?
A1. This is a good point that requires clarification. The images on Fig.2 intend to demonstrate an obvious change in the numbers of bacteria surviving antibiotic treatment supplemented with tRA99020 compared to no tRA addition to the antibiotic media. Therefore, we have chosen to compare the images at equal dilution rates which show higher CFU density in the absence of tRA99020. However, the calculation of CFU/ml used for the bar graph from 4 independent treatment experiments were based on dilution rates (usually higher than the ones shown on the graph) that provided optimum separation of the bacterial colonies. To clarify this, we added a sentence in the Figure 2 Legend (lines 241-243) explaining how the calculations were done: “For each experiment, the CFU/ml was determined from the agar plate with dilution rate resulting in well separated individual colonies (10-100 CFU) in three technical replicates.”
To address the Reviewer’s observation that we have used tRA20 compound at 1uM for subsequent experiments (Fig.2), we added a commentary in the Results’ section (lines 208-212) to clarify that the 1uM concentration had a varied effect on several Gram-negative bacterial species: “For these bacterial species, the reverse effect of tRA99020 on antibiotic bactericidal capacity was observed between 10 and 20 µM (data not shown)… and emphasize the differences in sensitivity towards the tRA99020 compound among bacterial species.”
- Figure 2 should be separated into four panels.
A2. Since the figure was already built from 4 panels, we assumed the Reviewer requested labeling each panel from A-D. This change was also captured in the legend of Figure 2.
- Figure 3B has no information about the statistical significance (p-values).
A3. We added to the figure 3 legend: “Notches on the boxplots describe the 95 % confidence interval and non-overlapping notches provide evidence of statistically significant difference of medians (p < 0.05, Sidak-Bonferroni method for multiple comparisons)”.
- Label-Free and single cell proteomics are the leading techiques in analyzing antibiotic resistance (Expert Review of Proteomics, 2019, 16(10):829-839), and recent technical advances can be discussed (Analytical Chemistry, 2022, 94(15):6026-6035; Microsystems & Nanoengineering, 2022, 8:13).
A4. We enhanced the discussion on the application of proteomics in studies of antimicrobial tolerance and resistance (lines 548-555) and added some of the suggested citations (ref # 49-53).
- Authors of reference #34 is not anonymous, check carefully.
A5. This has been corrected. Thank you!
- The layout of Figure 6 should be rearranged to avoid large blank.
A6. The layout of this figure has been changed to improve the quality of data presentation.
- Check the manuscript carefully for redundant spaces between words.
A7. We performed a thorough check of the spelling and have removed the extra space between the words whenever we were able to detect such. Thank you!

Reviewer 2 Report
In this work, the authors have investigated the role of a non-toxic taxane reversable agent, a small molecule tRA99020 that was initially developed to treat human cancer cell line by potentially blocking the ABC transport system to treat Burkholderia thailandensis persistor population. Previously the authors have identified Burkholderia thailandensis persistor population showed increased expression of polyketide synthesis system thereby aiding in lipid remodeling. The authors found out that treatment with this small molecule leads to decreases efflux pump capacity as shown by greater retention of the fluorescent dyes. Interestingly upon treatment with antibiotic and tRA099020, the exponentially dividing bacterial populations showed statistically significant decrease in viability and 15% decrease in the expression of pks operon.
Previous studies by authors have shown that the metabolites of pks operon can inactivate the mammalian Histone Diacetylase that aids in host immune response to the pathogen. The authors have tested the level of acetylated H3 in PMBC cell line treated with the pathogen in presence and absence of antibiotic. The authors found that untreated cells where able to activate the H3 acetylation whereas antibiotic treatment led to decrease in the bacterial population thereby reducing the acetylation of H3. Finally, the authors showed that antibiotic treatment exacerbates the cytotoxic effect from Burkholderia natural products 365 and inhibition of bacterial efflux capacity with tRA99020 counteracts this effect by restoring CFLAR 366 gene expression in host cells.
Overall, the work is solid. The paper is well written, and the materials and methods are easy to follow. It should be of interest to broader audience involved in studying host-pathogen interactions. I’ve only couple of comments regarding that I’m addressing below:
1. In Figure 1B, there was lack of dose dependent increase in RFU for tRA99020. The authors should repeat the experiment and show dose dependent response.
2. 2. It was not clear from the paper that if the effect of tRA099020 on pks and fadE is direct or indirect by activating stress response pathways which in turn can change the expression level of these two genes. The authors should discuss on that.
Author Response
Reviewer 2
Overall, the work is solid. The paper is well written, and the materials and methods are easy to follow. It should be of interest to broader audience involved in studying host-pathogen interactions. I’ve only couple of comments regarding that I’m addressing below:
- In Figure 1B, there was lack of dose dependent increase in RFU for tRA99020. The authors should repeat the experiment and show dose dependent response.
A1. We did investigate dose dependent inhibition of bacterial efflux capacity (visualized as increase in RFU) and did not find any. It may be the case that the method used is not sensitive enough to detect tRA99020 dose dependent retention of the fluorescent dye. Another possibility is that some drugs have distinct threshold effect below which there is no significant effect of the treatment and above it a fast saturation effect is observed. We added discussion of this observation in the Result section, lines 184-186.
- It was not clear from the paper that if the effect of tRA099020 on pks and fadE is direct or indirect by activating stress response pathways which in turn can change the expression level of these two genes. The authors should discuss on that.
Since tRA99020 does not affect gene transcription in general (Fig3C, fade transcription remained unchanged by tRA99020 and qPCR data not shown here has demonstrated no change in the transcript levels of housekeeping genes when bacteria were treated with tRA99020), we believe that the small molecule compound activates persistence gene expression indirectly by creating stressful conditions in the bacterial cells due to disruption of metabolite homeostasis streaming from inhibition of bacterial efflux capacity.
Following the Reviewer’s suggestion, we added couple sentences in the Results (lines 319-322) and in the Discussion section (lines 520-523) to address this important question.

Round 2
Reviewer 1 Report
The comments are addressed properly. The manuscript is good for publication.
Reviewer 2 Report
The authors have clarified all the points and is good for publication.